# Preparation and Optimization of Water-Soluble Cationic Sago Starch with a High Degree of Substitution Using Response Surface Methodology

**DOI:** 10.3390/polym12112614

**Published:** 2020-11-06

**Authors:** Nur’Izzah Md Nasir, Emilia Abdulmalek, Norhazlin Zainuddin

**Affiliations:** Department of Chemistry, Faculty of Science, Universiti Putra Malaysia, 43400 UPM Serdang, Selangor Darul Ehsan, Malaysia; nurizzahmdnasir@gmail.com (N.M.N.); emilia@upm.edu.my (E.A.)

**Keywords:** cationic sago starch, response surface methodology, high degree substitution, water-soluble starch

## Abstract

Modification and characterizations of cationic sago starch with 3-chloro-2-hydroxypropyl trimethylammonium chloride (CHPTAC) prepared via etherification reaction was reported in this study. The optimization of cationic sago starch modification was performed by utilizing the combination of response surface methodology and central composite design (RSM/CCD). The effect of each variable and the interaction between the three variables, the concentration of CHPTAC, concentration of the catalyst NaOH, and the reaction times on the degree of substitution (DS) of the product were investigated and modeled. Moderate conditions were employed and a water-soluble cationic sago starch with high DS value was obtained. Based on RSM, the highest DS = 1.195 was obtained at optimum conditions: 0.615 mol of CHPTAC concentration (CHPTAC/SS = 5), 30% *w*/*v* NaOH, and 5 h reaction time, at 60 °C reaction temperature. Furthermore, the cationic sago starch was characterized using Fourier transform infrared spectroscopy, FTIR, X-ray diffraction, XRD, and field emission scanning electron microscopy, FESEM.

## 1. Introduction

Cationic starches are important industrial derivatives in which the starch is given positive ionic charge by introducing ammonium, amino, imino, or phosphonium groups [1]. Utilization of cationic starches mainly located in the field of papermaking [2,3,4,5,6], as additives in manufacturing textile [7,8,9] and cosmetics [10], or flocculants in wastewater treatment [11,12]. In addition, studies on the application of cationic starch in drug delivery, are also in progress [13,14,15]. Many research papers and even patents according to Hebeish et al. [8], Jelkmann et al. [14], Putro et al. [15], Khalil and Aly [16], Siau et al. [17], Heinze et al. [18], Bratskaya et al. [19], Kuo and Lai [20], Liu et al. [21], and Cheng et al. [22] have been published on the preparation, manufacturing, and application of cationic starch. Among the cationic starch preparation, the interest on cationic starch with a high degree of substitution (DS) is increasing owing to the promising candidate as improvise additive or other wide application.

In previous studies, most of the researchers used starch from maize, waxy maize, and potato for cationic modification. Different approaches were done to obtain the high DS value cationic starch. Conventionally, the cationic starch can be prepared by the etherifying [8,18], graft polymerization [11], or esterification [23] reactions of starch with the tertiary amine and quaternary ammonium salt reagents, such as glycidyl trimethylammonium chloride (GTAC), 3-chloro-2-hydroxypropyl trimethylammonium chloride (CHPTAC) or 2,3-epoxypropyl trimethylammonium chloride (EPTAC) in heterogeneous or homogeneous reaction conditions. A study from Heinze et al. [18] has used both heterogeneous and homogeneous conditions to produce high DS cationic starch with up to DS = 1.05 in different reaction media such as ethanol/water, dimethylsulfoxide (DMSO) or under the dissolution of NaOH solution. A similar study by Hebeish et al. [8] but using a variety of catalysis and they obtained DS value up to 0.85 at 70 °C after 3 h reactions. Green chemistry approach by employing ionic liquid 1-butyl-3-methylimidazolium chloride as cationization solvent resulting DS up to 0.99, after 2 h at 80 °C [24]. Another approach via graft polymerization of starch performed by Anthony [11] also resulting in a high DS product with 0.83 at 75–80 °C. Esterification reaction also can be used to produce cationic starch by conversion waxy maize starch with CHPTAC after activation with N,N´-carbonyldiimidazole at 70 °C also produced high DS with 0.73 [23]. However, the choice of the reaction temperature is very important because a high temperature might cause protein denaturing in starch structure [25]. Protein begins to undergo thermal denaturation when the temperature is about 60–70 °C which can exhibit loss of solubility to aggregation [26,27]. Therefore, we need to avoid denaturing of protein to produce water-soluble cationic starch.

Sago starch is another type of starches that are used in the food industry and selected as the raw material because it is relatively cheap and widely available in Southeast Asia especially Malaysia and Indonesia. Sago can compete economically in terms of yield and price compared with other starch crops [17,28]. Even though the studies on cationization of starch with high DS value had widely been done, only a few studies had been used sago starch and most of them did not produce high DS value. For example, Siau et al. [17] has prepared cationic starch by dissolution in an alkaline solution using the RSM approach and the DS obtained was only 0.06.

Response surface methodology (RSM) is one of the popular designs of experiment techniques. This method implements statistical and mathematical techniques for the development, improvement and optimizing process, in which the response of interest is affected by several variables. This statistical process works by defining the effect of independent variables, individually or in combination, thus allows the determination of the significance of each parameter studied, along with the significant interaction between parameters. Central composite design (CCD), Doehlert design (DD) and Box–Behnken design (BBD) are among the typical designs in RSM [29]. Currently, limited reports are available on RSM for optimization in high DS cationic starch. In this study, a central composite design was implemented to determine the changes in DS value and reaction efficiency of the cationic sago starch under the varying concentration of sago starch, reaction time, and amount of cationic reagent. The heterogeneous method approach was applied in this study. The reaction was in moderate conditions where low concentration and toxicity of chemicals were used as well as the experiment was easy to handle. Morphology and molecular properties of cationic sago starch were characterized to verify the deduction according to the RSM analysis.

## 2. Materials and Methods

### 2.1. Materials

Sago starch (food grade) was obtained from C.L Nee Sago Industries Sdn Bhd, Sarawak, Malaysia. The cationic etherifying agent, 3-chloro-2-hydroxypropyl trimethylammonium chloride (CHPTAC) with 60% (*w/v*) activation monomer concentration, methanol (99.7%), isopropanol (99.7%), and ethanol (95%) were purchased from Sigma-Aldrich (M) Sdn. Bhd., Kuala Lumpur, Malaysia. 85% (*v/v*) methanol was prepared from methanol (99.7%). All other chemicals such as isopropanol (99.7%), and ethanol (95%) were of analytical grade and used as received without further purification.

### 2.2. Preparation of Cationic Sago Starch

The method from Heinze et al. [18] was modified to prepare cationic sago starch. Dried sago starch, SS (10 g, 0.123 mol) was suspended in 80 mL isopropanol (99.7%). Sodium hydroxide (30–70% *w/v*) was dissolved in 20 mL water and added in the starch suspension. Subsequently, a solution of CHPTAC (0.123–0.615 mol), with molar ratio CHPTAC/SS = 1 to 5 was added dropwise into the suspension. The mixture was kept at 60 °C (3–5 h). After cooling down to ambient temperature, the product was neutralized with 1 M hydrochloric acid, filtered off and finally washed with ethanol (95%).

### 2.3. Design of Experiment and Response Surface Methodology

Important variables on the cationization of sago starch with CHPTAC were optimized by using the response surface methodology/central composite design (RSM/CCD) approach. CHPTAC concentration (0.123–0.615 mol), concentration of NaOH (30–70%), and reaction time (3–5 h) were the independent variables used in this study. Three factors with a total of 16 experiment runs were suggested by RSM/CCD using the statistical package from Design Expert 6.0, Stat Ease Inc., Minneapolis, MN, USA.

Response surface modeling, statistical analysis, and optimization were conducted with the assistance of Design Expert 6.0. Analysis of variance (ANOVA) was used in the analysis of the output data. The degree of substitution (DS value) data from the suggested experiments was expressed by polynomial regression equation to generate a model as shown in the equation below:$$Y=\beta_{0}+\sum_{i=1}^{k}\beta_{ii}x_{i}+\sum_{i=1}^{k}\beta_{ii}x_{i}{}^{2}+\sum_{i=1}^{k-1}\sum_{j>i}^{k}\beta_{ij}x_{j}$$
where Y represents the predicted responses, $\beta_{0}$ is the constant coefficient, $\beta_{i}$ is the *i*th linear coefficient, $\beta_{ii}$ is the quadratic coefficient, $\beta_{ij}$ is the *ij*th interaction coefficient, while *x_i_* and *x_j_* represents factors. Validation of the model and optimization of the degree of substitution response were conducted. The optimum values of variables were predicted by the response surface analysis of the combined variables. The constraints in this study were applied to predict the optimum variable conditions that resulting in the highest degree of substitutions value response by inserting the desired conditions in the Design Expert 6.0 software. The efficiency of the model was verified by performing the suggested experiments and comparing the output with the predicted results. Residual standard errors (RSE) were used to determine the reliability of the model. The numerical optimization of the response was predicted based on the second-order polynomial model.

### 2.4. Determination of DS Value of Cationic Sago Starch

The degree of substitution (DS) is the number of hydroxyl groups substituted per glucose unit with anhydrous glucose unit, AGU. The DS was determined by elemental analysis of a TruSpec Micro CHNS-932 Analyzer, St. Joseph, Michigan, MI, USA. To determine the percentage of nitrogen present. Five replicates were analyzed with a low standard deviation (<0.5). The DS value can be calculated using the following equation [24]:$$DS=\frac{162N}{\left(1400-CA\times N\right)}$$
where, 162 = molecular weight of anhydrous glucose unit, AGU; *N* = percentage of nitrogen; and *CA* = molecular weight of cationic reagent.

### 2.5. Characterization of Cationic Sago Starch

Cationic sago starch were characterized using Fourier transform infrared spectroscopy 100 series Perkin Elmer (Waltham, MA, USA), X-ray diffraction spectroscopy XRD-6000 (Shimadzu, Kyoto, Japan), and field emission scanning electron microscopy JSM-7600F (JEOL, Tokyo, Japan) 

Fourier transform infrared, FTIR measurements were carried out with 3 mg of sample on the diamond holder was irradiated with infrared in the range of 4000 to 400 cm^−1^ using attenuated total reflectance (UTAR) technique.

X-ray Diffraction, XRD characterization was carried out using Shimadzu XRD-6000 diffractometer with Cu Kα (λ  =  1.5418 Å) radiation at room temperature operated at 30 kV and 30 mA. A sample was placed in an aluminum sample holder and a diffraction pattern plots intensity against the angle of the detector, 2θ and the scanning range 2° to 30° with the rate of 2°/min with continuous scan mode.

Field emission scanning electron microscopy (FESEM) was used for surface morphology studies. Samples were mounted on the stub followed by coating with gold to make the samples conductive before the scanning process is done. The prepared samples were examined under FESEM (JEOL, JSM-7600F) at a voltage of 5.0 kV and recorded at 200 and 1000× magnification.

## 3. Results and Discussion

### 3.1. Data Analysis and Evaluation of the Model

#### 3.1.1. Development of Regression Model Equation

The complete design matrix for preparing cationic sago starch is given in Table 1. The DS experimental obtain was in the range of 0.45 to 1.1. For this response, a quadratic model was suggested according to the sequential model sum of squares. The final empirical formula model for the DS in terms of a coded factor was shown in the following equation:$$\begin{array}{l} \sqrt{DS}=0.75+0.11A-0.098B-(8.402\times 10^{-4})C-0.096A^{2}-0.077B^{2}\\ +0.11C^{2}-0.092AB+0.042AC-0.063BC \end{array}$$
where, *A* = concentration of CHPTAC, *B* = concentration of NaOH, and *C* = reaction of time. The quality of the model was evaluated based on correlated coefficient and standard deviation. Model capability to predict the response was translated to a small deviation and correlated coefficient closer to the R^2^ [30]. The standard deviation and correlated coefficient for this model were 0.039 and 0.9772, respectively. When the correlated coefficient almost approaches 1, this implied that there was a good fit between the actual values and the values predicted by the network [31]. These indicate that 97.72% of the total variation in the DS was attributed to the experimental variables studied. These values were considered high as the value were close to unity and lead to a small variation in DS predicted.

#### 3.1.2. Analysis of Variance, ANOVA

The ANOVA was determined by using Design Expert 6.0 software. The statistical significance of this model was determined by F-value and prob > F. F-value is a measurement of the variance of data about the mean, based on the ratio of the mean square of group variance due to error. The model is significant when it possesses high F-value and prob. > F less than 0.05. The ANOVA for the quadratic model of the DS value is presented in Table 2. The F-value and prob F > for this model were 28.57 and 0.0003, respectively, indicating this model was significant. In case A, B, A2, B2, C2, AB, AC, and BC were significant model terms and only C was insignificant model terms to the response. Meanwhile, the lack of fit implies the insignificant value which is relative to pure error as the prob > F value is 0.0795, which is greater than 0.05. Not a significant lack of fit indicates that this model has good predictability. Based on the statistical results obtained, the model adequately predicted the DS value within the range of variables studied. Figure 1 shows the predicted value versus the experimental values. The response model showed good fits to experimental data, demonstrating good predictions of the model.

#### 3.1.3. Effect of Independent Variables on the Values of DS

The cationization starch involving several steps of reactions simultaneously. First, the chlorohydrin form of the CHPTAC is converted to epoxy intermediate. Next, the epoxy agent will react with the starch itself in the presence of an alkaline medium. Then, the epoxy unit will be converted via hydrolysis to the non-reactive 2,3 dihydroxyl derivative [8]. The illustration reaction scheme is shown in Figure 2.

##### Effect of CHPTAC and NaOH Concentrations on DS

The effect of CHPTAC and NaOH concentrations on the value of DS with a constant reaction time at 4 h is illustrated in Figure 3a. It can be seen that the DS value of cationic starch increased gradually with an increase in the CHPTAC concentration molar ratio up to 0.369. This trend can be explained as increasing in CHPTAC concentration, the possibility of an etherification reaction between sago starch and CHPTAC molecule also increases. The higher the concentration of CHPTAC will increase the availability of the CHPTAC to fill up the vicinity of the hydroxyl group in sago starch due to the epoxidation formation. However, by increasing the concentration of CHPTAC beyond 0.369, the DS value was almost invariable. Therefore, the effect of the excess amount of CHPTAC on the cationization reaction was not significant under the given reaction conditions. It can be that the optimum CHPTAC concentration for the reaction was 0.369. A similar result was obtained by Wang and Xie [24]. The excess of cationization reagent at certain optimum conditions does not disturb the etherification reaction causes the DS to remain constant. However, for the effect of NaOH concentration, the DS showed the highest DS at 30% NaOH concentration and start to decrease with the increase of the concentration of NaOH. Therefore, we can conclude that the concentration of CHPTAC and NaOH are significant factors influencing the value of DS. 

##### Effect of CHPTAC Concentration and Reaction Time on DS

Figure 3b shows the effect of CHPTAC concentration in the range of 0.123 to 0.615 mol and reaction time in the range of 3–5 h. The effect of concentration of CHPTAC on DS can be clearly seen on this figure as the DS increases with CHPTAC concentration. Nevertheless, there is only little effect of reaction time on DS. This is in good agreement with Zainal Abiddin et al. [33], which stated that long reaction times do not necessarily produce high DS value because anhydride needed for cationization is exhausted, thus hydrolysis will take place. Even though dissolution and diffusion of CHPTAC into starch increase with increasing reaction time, at one point, side reaction will compete with etherification in the production of cationic sago starch. Finally, the side reactions become dominant as prolonging the reaction time and concentration of CHPTAC was depleted due to etherification and hydrolysis [34].

##### Effect of NaOH Concentration and Reaction Time on DS

In the formation of cationic sago starch, NaOH acts as a catalyst in both reactions which are the formation of intermediate epoxide along with cationization of the starch [8]. From Figure 3c, the highest DS was achieved at NaOH concentration around 30%. The addition of NaOH until 30% *w*/*w* of concentration increased the DS rapidly. However, the DS then gradually decreased after 30% *w*/*w* NaOH due to hydrolysis. This phenomenon can be explained that under the basic condition the hydroxyl groups of the anhydroglucose units of starch may react in a nucleophilic reaction with the etherification agent, as in the well-known Williamson ether synthesis [35]. But at high alkali concentration, the cationization agent undergoes a hydrolysis reaction, which affects an epoxide ring-opening to form the diol product. A similar trend is also reported by Wang and Xie [24] who studied the modification of cationic starch with high DS using ionic liquid. Hence, relatively low NaOH concentration (30%) is more suitable for substitution reaction to reduce undesirable secondary reactions.

#### 3.1.4. Validation of Model and Optimization of DS Value

Validation of the model was carried out by performing three sets of experiments generated from Design Expert 6.0 software. The validation was obtained by comparing the predicted and experimental values. The optimization process was carried out by inserting the desirable criteria such as the maximum DS value response was set as the main goal, while the other factors were in the studied range as mention in methodology Section 2.3. The experiments and the response from the optimization of DS value in Table 3 indicated that all the experiments produce DS values response less than 2% residual standard error (RSE), thus validating the model. Based on the performed experiments, the prediction of this model is accurate up to 98%. Experiment 1 in Table 3 was chosen as an optimum condition according to RSM software. Even though experiment 2 gave the highest DS value, the RSE% was a little bit higher than experiment 1. Interestingly, the DS value obtained by the suggestion experiment from RSM gave slightly higher DS with 1.195. According to Pfeifer et al. [23], water-soluble starch can be obtained from the cationic starch with DS more than 1.

### 3.2. Cationic Sago Starch Characterization

#### 3.2.1. Fourier Transform Infrared, FTIR Spectroscopy Analysis

FTIR spectra of sago starch, CHPTAC, and cationic sago starch are depicted in Figure 4. The sago starch spectrum in Figure 4a shows common absorption bands for polysaccharides with glucopyranose rings such as the O-H band at 3305 cm^−1^, C-H stretching vibrations of aliphatic groups at 2906 cm^−1^, absorbed water signals at 1644 cm^−1^, C-H bending at 1365 cm^−1^, C-O stretching for ether at 1000 cm^−1^. The band around 1644 cm^−1^ corresponded to the tightly bound water present in starch due to its hygroscopic nature. In general, starch granules have two different populations of water, one that is part of the hydrogen bonding network of the starch crystalline structure and another which is freer and considered hydration water [36].

IR spectrum for CHPTAC in Figure 4b shows a broad absorption band at 3337 cm^−1^ which is assigned to the O–H stretching vibration of water since the CHPTAC is 60% *w*/*v* in liquid form. A strong band was observed at 1649 cm^−1^ due to intermolecular H-bond in the CHPTAC. Another strong band at 1478 cm^−1^ corresponded to C–N stretching vibration. There are two strong absorption bands at 944 cm^−1^ and 1081 cm^−1^ corresponded to an alkyl halide, C–Cl and C–O stretching, respectively.

Figure 4c represents the IR spectrum of cationic sago starch at the optimum condition with DS 1.195. The IR spectrum of cationic sago starch has similar absorption bands as unmodified sago starch. This is because the backbone of starch itself is large and only one out of three hydroxyl groups in the starch molecule were substituted with CHPTAC. Theoretically, the highest possible DS for cationic starch is 3 since the anhydroglucose unit of starch provides three active hydroxyl groups, including one primary and two secondary hydroxyl groups [22,37]. A possible chemical scheme for the formation of cationic sago starch is shown in Figure 5. An additional absorption band at 1445 cm^−1^ attributed to the C–N stretching vibration of amide, which similar to Xia et al. [38] observed. This C–N band is evidence of the incorporation of cationic moiety onto the backbone of sago starch [8].

#### 3.2.2. X-ray Diffraction, XRD Analysis

Starch is a semicrystalline material which contains both the crystalline and amorphous phases, determined by its principal components, the amylose and the amylopectin. The amylose has a linear structure, which produces the crystalline region of the starch, whereas the amylopectin has a branched structure that is responsible for the amorphous phase of the starch [39,40]. The existence of two phases is confirmed by the XRD pattern. The diffractogram for sago starch is shown in Figure 6a. The main peaks were observed at 2θ=14.96°,16.9°,18.04° and 23.16°. A broad peak with low intensity is observed between 19.2° and 21.5°. Similar results were reported by Todica et al. [41] for corn starch and Tuan et al. [42] for sago starch. X-ray diffraction pattern for cationic sago starch is shown in Figure 6b. The diffractogram for cationic starch only had a dispersive broad peak and showed no crystal peak like unmodified sago starch. High DS cationic starch prepared by Wang and Xie [24] also observed a similar broad peak in the XRD pattern. This implies that the crystallinity of starch has been damaged completely during the cationization reaction. This loss in crystallinity might due to the replacement of the hydroxyl group in starch with the cationic group during the modification process [43].

#### 3.2.3. Field Emission Scanning Electron Microscopy, FESEM Analysis

Figure 7a,b shows the FESEM micrographs of the sago starch and cationic sago starch at 200× and 1000× magnification. FESEM revealed that sago starch consists of oval granules with smooth surfaces and some curtailed side [42,44]. When cationization takes place, the crystallinity in the sago starch molecule starts to lose and disintegrate. This disintegration of the granule shape of sago starch is shown in Figure 7b. High DS causes the surfaces of the starch granules to completely disintegrate, and their edges drastically lose definition. Apparently, cationization destroys the structure of the sago granule and reduces hydrogen bonding. These factors facilitate rapid water uptake to the point that the high DS cationic starch will dissolve in water [35]. 

## 4. Conclusions

In this work, the central composite design/response surface methodology is successfully utilized for the optimization of DS in the cationization of sago starch. The effect of three major factors in the cationization of sago starch was studied and optimized statistically. Based on RSM, the optimum conditions to prepare water-soluble cationic sago starch in moderate conditions were at 0.615 mol of CHPTAC concentration (CHPTAC/SS = 5), 30% *w/v* NaOH, 5 h reaction time, and 60 °C reaction temperature. The value of DS obtained from these optimum conditions was 1.195. The validation and optimization based on the software prediction have shown that this model can predict with less than 2% error as compared with the experiment, suggesting that this model was reliable and able to predict the DS values response accurately.

## Figures and Tables

**Figure 1 polymers-12-02614-f001:**
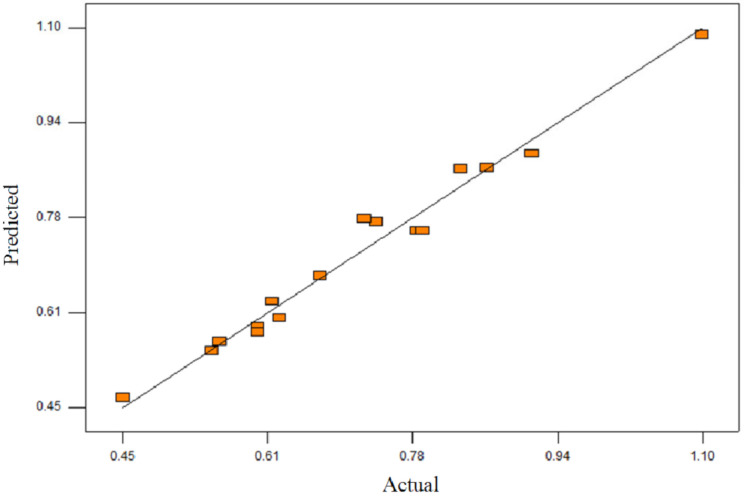
The plot of predicted versus actual (correlation between the predicted and actual experiment values for the DS values).

**Figure 2 polymers-12-02614-f002:**
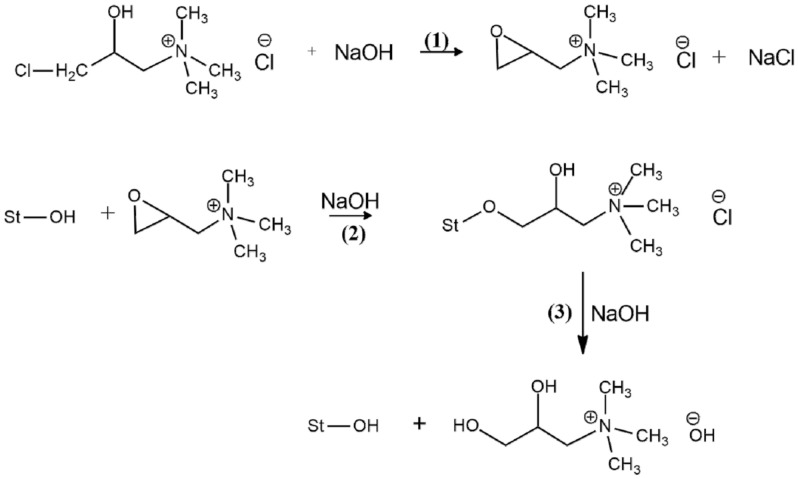
Illustration cationization starch chemical scheme reactions, (**1**) epoxidation formation, (**2**) cationic starch formation, and (**3**) hydrolysis reaction with an excess of NaOH [32].

**Figure 3 polymers-12-02614-f003:**
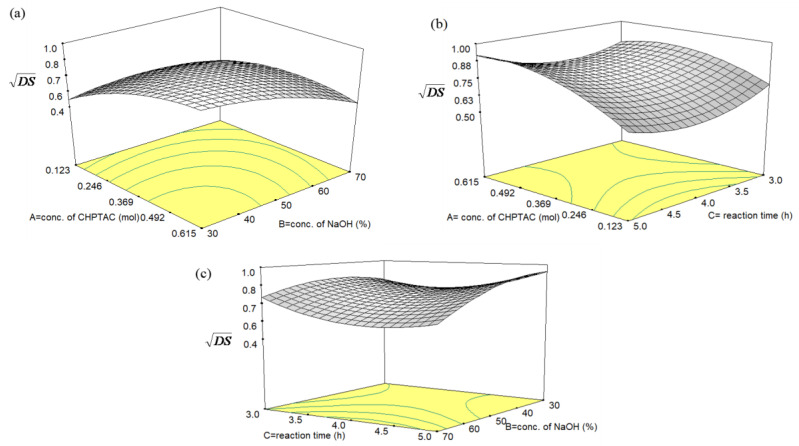
Response surface curve for effects of (**a**) CHPTAC conc. (mol) and NaOH conc. (%), (**b**) CHPTAC conc. (mol) and reaction time (h), (**c**) NaOH conc. (%) and reaction time (h) on the value of DS.

**Figure 4 polymers-12-02614-f004:**
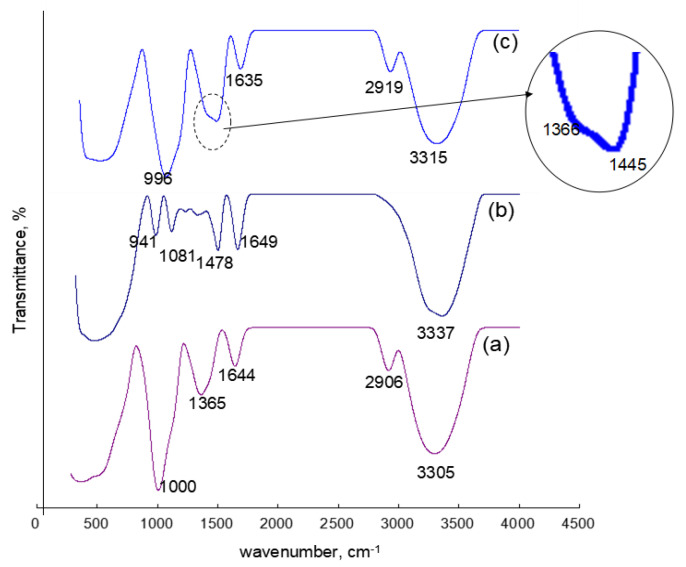
FTIR spectra for (**a**) sago starch, (**b**) CHPTAC, (**c**) cationic sago starch with DS = 1.195 at wave number 0–4000 cm^−1^.

**Figure 5 polymers-12-02614-f005:**
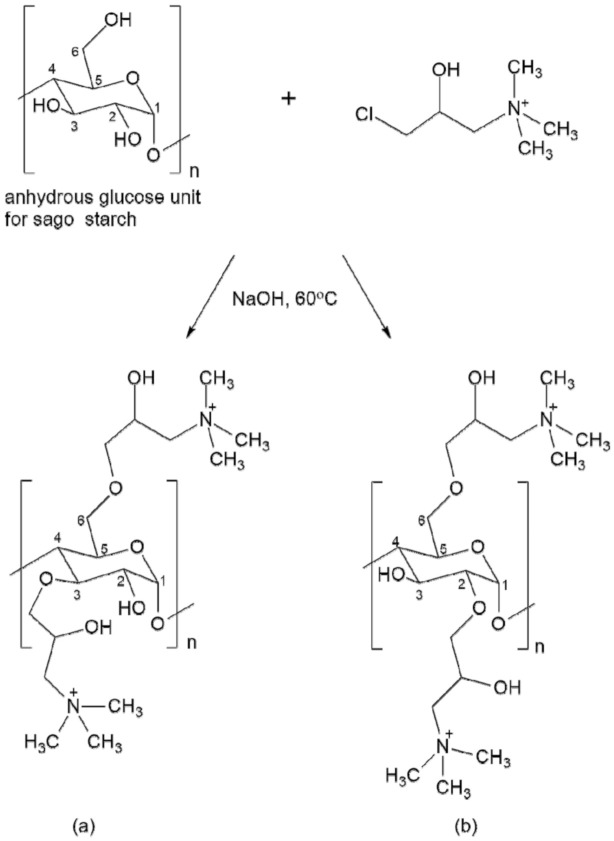
Illustration of possible cationic structures for one-unit anhydrous glucose in sago starch (DS = 1.195); (**a**) C_3_-O and C_6_-O attachments, and (**b**) C_2_-O and C_6_-O attachments.

**Figure 6 polymers-12-02614-f006:**
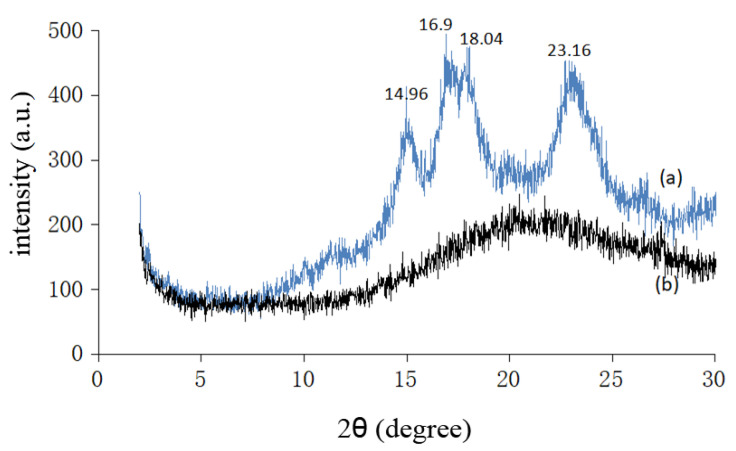
XRD patterns for (**a**) sago starch, and (**b**) cationic sago starch with DS = 1.195.

**Figure 7 polymers-12-02614-f007:**
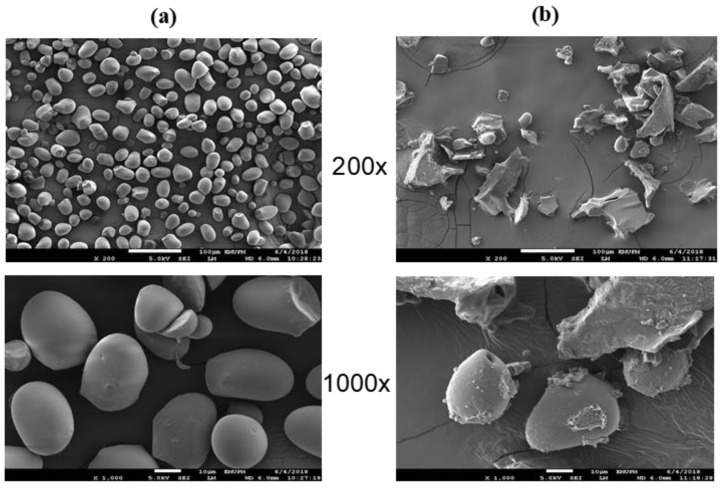
Field emission scanning electron micrograph of (**a**) sago starch, and (**b**) cationic sago starch with DS 1.195 at ×200 and ×1000 magnification.

**Table 1 polymers-12-02614-t001:** Central composite design (CCD) for cationization of sago starch experimental degree of substitution response data.

Variables				Response (Experimental)
Sample	A: Conc. of CHPTAC (mol)	B: Conc. of NaOH (% *w*/*v*)	C: Reaction Time (h)	Degree of Substitution, DS
S1	0.123	30.0	3.00	0.56
S2	0.615	30.0	3.00	0.91
S3	0.123	70.0	3.00	0.67
S4	0.615	70.0	3.00	0.62
S5	0.123	30.0	5.00	0.62
S6	0.615	30.0	5.00	1.10
S7	0.123	70.0	5.00	0.45
S8	0.615	70.0	5.00	0.60
S9	0.123	50.0	4.00	0.55
S10	0.615	50.0	4.00	0.73
S11	0.369	30.0	4.00	0.72
S12	0.369	70.0	4.00	0.60
S13	0.369	50.0	3.00	0.86
S14	0.369	50.0	5.00	0.83
S15	0.369	50.0	4.00	0.78
S16	0.369	50.0	4.00	0.79

**Table 2 polymers-12-02614-t002:** ANOVA of the degree of substitution, degree of substitution (DS) for response surface quadratic model.

Source	Sum of Squares	D.O.F	Mean Square	F Value	Prob > F
Model	0.396	9	0.044	28.57	0.0003
A	0.125	1	0.125	81.35	0.0001
B	0.097	1	0.096	62.75	0.0002
C	7.06 × 10^−6^	1	7.06 × 10^−6^	0.46 × 10^−2^	* 0.9482
A^2^	0.025	1	0.025	15.88	0.0072
B^2^	0.016	1	0.016	10.16	0.0189
C^2^	0.031	1	0.031	19.88	0.0043
AB	0.068	1	0.068	43.91	0.0006
AC	0.014	1	0.014	8.99	0.0241
BC	0.031	1	0.031	20.38	0.0040
Lack of Fit	0.92 × 10^−2^	5	0.18 × 10^−2^	90.81	* 0.0795
R^2^	0.9772				
Adjusted R^2^	0.9430				
Predicted R^2^	0.8178				
Adeq. precision	20.272				
STD	0.039				

* not significant; D.O.F = degree of freedom; STD = standard deviation.

**Table 3 polymers-12-02614-t003:** Predicted and observed response values conducted at an optimum combination.

Exp.	Conc. of CHPTAC (mol)	Conc. Of NaOH (%*w*/*v*)	Reaction Time (h)	$\sqrt{DS}$		Residue Standard Error, RSE%	Reaction Efficiency, RE%
				Predicted	Experiment		
* 1	0.615	30.00	5.00	1.093	1.0936	0.05	100
2	0.600	30.00	4.99	1.084	1.1027	1.73	100
3	0.530	37.40	3.00	0.896	0.888	0.89	99.1

* Choose as the optimum condition.

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
