# Peer review of "Preparation and Optimization of Water-Soluble Cationic Sago Starch with a High Degree of Substitution Using Response Surface Methodology"

_polymers, 2020, doi:10.3390/polym12112614_

Round 1

Reviewer 1 Report

The manuscript “Preparation and optimization of water-soluble cationic sago starch with a high degree of substitution using response surface methodology” [polymers-963495-peer-review-v1] written by Nur’Izzah Md Nasir, Emilia Abd Malek and Norhazlin Zainuddin describes a modification of sago starch with 3-chloro-2-hydroxypropyl trimethylammonium chloride to a cationic sago starch. The reaction was optimized with respect to concentration of 3-chloro-2-hydroxypropyl trimethylammonium chloride (CHPTAC), NaOH concentration (catalyst), and the reaction time using combination of central composite design and response surface methodology. The produced cationic sago starch was identified by FT IR, X-ray, XRD and FESEM.

The scientific work is well planned and performed. All the reactions and investigations have been made carefully and only state of the art methods have been used. The discussion is clear and all conclusions sound sensible and detailed with respect to entire data gained. The reported results sound perspicuous. The results are hence of some interest in the fields of Polymer Chemistry, Organic Chemistry and Carbohydrate Chemistry.
However, there are a few comments listed below, which should be taken into account by the authors prior to acceptance of the manuscript.

Comments:

a) Figure 2 and 4: The chemical structures should be drawn with uniform bond lengths and the bond angles customary for such structural formulas.

b) Figure 4: The glycol structures shown are D-altrose (in the upper left and in a) and D-allose (in b). The authors are urged to change the position of the OH and OR groups so that they represent D-glucose. Also the interglycosidic linkages should be shown in alpha, and somewhere between alpha and beta.

c) The authors are encouraged to give units in the SI units form. E.g. Line 93: h instead of "hours

Author Response

Response to Reviewer 1 Comments

Point 1: Figure 2 and 4. The chemical structure should be drawn with uniform bond lengths and the bond angles customary for such structural formulas

Response 1: The uniform bond length had been corrected at figure 2 and 5 in lines 187 and 277 respectively

Point 2: Figure 4; the glycol structures shown are D-altrose (in the upper left) and in (a), and D-allose in (b). The authors are urged to change the position of OH and OR groups so that they represent D-glucose. Also the interglycosidic linkages should be shown in alpha, and somewhere between alpha and beta.

Response 2: The glycol structure that the reviewer mentioned is actually in figure 5. The glycol structure had been corrected and the position of OH and OR had been changed. The OH attachment has been corrected into equatorial position to represent the D-glucose structure, and the interglycosidic linkages have been corrected in alpha position in figure 5 upper left, (a), and (b) in line 277.

Point 3: The authors are encouraged to give units in the SI units form. E.g. Line 93: h instead of hours.

Response 3: The changes have been made as suggested by the reviewer. The changes were made for Mole=mol ; and hours = h.

The changes can be seen in lines 19, 20, 49, 94, 100, 160(Table 1), 207-208(figure 3), 211, 246(Table 3), 317, and 318

Reviewer 2 Report

Recommendation: is accept after minor revision

Comments to Authors:  Nur’Izzah Md Nasir, Emilia Abd Malek  and Norhazlin Zainuddin

Manuscript Number: polymers-963495-peer-review-v1

Article Type: Article

Article Title:  Preparation and optimization of water-soluble cationic sago starch with a high degree of substitution using response surface methodology

Overview and general recommendation

The introduction provides a sufficient background and contains relevant references to the problem raised.

The methods were presented correctly and chronologically adequate to the conducted research and are adequately described but can be improved

The research design is appropriate.

Minor comments

  • line 18: please explain what mean mild conditions?
  • keywords: In my opinion word sago starch is not necessary
  • keywords: better is phrase: water-soluble starch, not only water-soluble
  • line 92: CHPTAC/SS = 1 -5, not CHPTAC:SS = 1:5?
  • lines 121- 122: information should be more accurate, e.g. CHNS-932 Elementary Chemical Analyzer (LECO, USA)
  • line 125: any source of bibliography for the degree of substitution (DS)? From where is equation for DS?
  • line 126-127: let me know if there should be some units
  • 3 not clear axes, lease correct
  • line 305, fig. 7: (…) at x200 and x1000 magnification is enough, without double word magnification

Overall Recommendation

My recommendation is accept after minor revision

Author Response

Response to Reviewer 2 Comments

Point 1: Line 18; please explain what mean mild conditions

Response 1: From our understanding, mild conditions are a simple and easily manageable experiment. In our studies, the method was simple which involved mixing using a water bath shaker and the sample was washed using alcohol. The temperature involved also low with only up to 60°C. The chemicals were also low in concentration and low toxicity. I have mention the mild condition in the introduction part line 77-79

Point 2: Keywords; In my opinion word sago starch is not necessary. Keywords; better is phrase, water-soluble starch, not only water-soluble

Response 2: The word ‘Sago starch’  in the keywords has been removed and the word ‘starch’ after water-soluble has been added in lines 24-25

Point 3: Line 92; CHPTAC/SS=1-5, not CHPTAC:SS = 1:5?

Response 3: It is CHPTAS/SS = 1-5 which means 1 to 5 molar ratio. We have changed it into ‘1 to 5’ instead of ‘1-5’ to make it more understandable in line 93.

Point 4: Line 121-122; information should be more accurate, eg; CHNS-932 Elementary Chemical Analyzer, (LECO, USA)

Response 4: The information has been added in line 122-123

Point 5: Line 125; any source of bibliography for the degree of substitution (DS)? From where is the eqn for DS?

Response 5: The citation has been added in line 125

Point 6: Line 126-127; let me know if there should be some units

Response 6: The degree of substitution does not have any unit because it represents the amount of possible cationic reagent react and replace the OH group from the glucose unit.

Point 7: Figure 3; not clear exes, lease correct

Response 7: We have changed into a clearer figure in Figure 3 in line 206

Point 8: Line 305; figure 7, at x200 and x1000 is enough,without double word magnification.

Response 8: The word magnification has been removed in Figure 7 in line 310.